# Developing Process for Selecting Research Techniques in Urban Planning and Urban Design with a PRISMA-Compliant Review

Abeer Elshater [1,*] and Hisham Abusaada [2]

1 Department of Urban Design and Planning, Faculty of Engineering, Ain Shams University, Cairo 11517, Egypt
2 Housing and Building National Research Center, Cairo 12622, Egypt
* Correspondence: abeer.elshater@eng.asu.edu.eg

**Abstract:** Choosing the proper research methods can pose a challenge for novice urban planners and designers. This study aimed to develop a more effective process for assisting urban planners and designers in selecting appropriate research techniques. The study used bibliometrics, systematic reviews, meta-analyses, and storytelling techniques as examples of urban planning and urban design research techniques. The results of this study provide techniques and procedures that can help urban planners and designers to conduct research reviews and follow the previous documented works published in the field. By utilizing suggested techniques and procedures, conclusive conclusions in urban planning and design research can be formed on the basis of compelling evidence. This study recommends developing a further innovative research methodology based on the Preferred Reporting Items for Systematic Reviews and Meta-Analyses by improving documentation and dissemination of research reviews.

**Keywords:** bibliometrics analyses; meta-analyses; research techniques; storytelling techniques; systematic reviews

## 1. Introduction

It is difficult for novice researchers to evaluate planning strategies, make theoretical contributions, and create new paradigms (Casanave and Li 2015), especially if they are not experienced in dealing with research methods and techniques (Chen et al. 2016). Employing research methods and techniques that rely on critical literature reviews and mapping new contributions has often resulted in discovering new knowledge (i.e., originality) (Baptista et al. 2015). Therefore, selecting appropriate research methods and techniques for analysis and criticism of relevant research remains a critical topic of research.

Novice researchers' inability to meet scientific writing standards is an important issue that should be addressed. Some researchers cannot formulate a specific research question (Dodgson 2020). Other researchers' works have essential methodological and structural defects, such as the failure to define research methods (Liu 2018), which hinders research flow based on goals, objectives, issues, contributions, and limitations. Determining the review contents and methods from the title of a study can also be challenging.

Despite extensive research on literature review techniques, urban planning and urban design have not adequately utilized changes in line with the techniques in the fields of social and medical sciences (Abusaada and Elshater 2022). Xiao and Watson (2019) believe that the field of urban planning lacks rigorous systematic reviews and knowledge of the results of theoretical analysis methods, which remain scattered and are only partially discussed in most academic studies. Fleming et al. (2014) added that Preferred Reporting Items for Systematic Reviews and Meta-Analyses (PRISMA) reports used in medical studies do not use meta-analysis of observational studies in epidemiology and quality assessment of the accuracy of diagnostic studies. However, urban planning and design research does not go through a similar process to PRISMA for medical studies of reviewing literature or

registering the research results (Abusaada and Elshater 2022; Xiao and Watson 2019). It is assumed that PRISMA is still not used in urban planning and design research in accordance with its specifications and protocols. In addition, the International Prospective Register of Systematic Reviews (PROSPERO), as an open access database documenting PRISMA health care results, is not for urban planning research. Using an irrelevant database such as PROSPERO in health care hinders the direction of cumulative outcomes in similar case studies conducted in urban planning and design. PRISMA is used in urban planning research to systematize the review of literature, with limitations for storing the cumulative outcomes in a specific database. Therefore, our understanding of establishing such a procedure of reviewing urban planning and research literature and documenting the results is also limited. We have also noted a gap in the relevant literature and effective use of research, such as a lack of systematic reviews in the postgraduate nursing education (Ham-Baloyi and Jordan 2016), urban planning (Abusaada and Elshater 2022), and appropriate academic research systems (Gusenbauer and Haddaway 2020).

This study aimed to develop a procedure to assist urban planners and designers in selecting appropriate research techniques. We examined four research techniques in this study: bibliometrics analysis, systematics, meta-analysis, and storytelling techniques (BSMS). This study used data mining techniques to search the databases of Scopus, Web of Science, and Google Scholar, focusing on quantitative and qualitative literature review methods. With improved documentation and dissemination, this study contributes to advancing urban planning and design research methodology based on PRISMA, which is not commonly used in the field.

After this introduction, this research is divided into three sections (Figure 1). First, we developed a three-stage research design and data collection methodology. The purpose of these stages was to answer the question of how a literature review fits into the search strategy for urban planning and design studies. As a result of the second section, 51 documents were examined using coding schemes, literature reviews, bibliometric analysis, systematic reviews, meta-analyses, and storytelling (BSMS). With guidance from the PRISMA statement, the third section of this research suggested 20 steps for conducting a literature review using BSMS in urban planning and design.

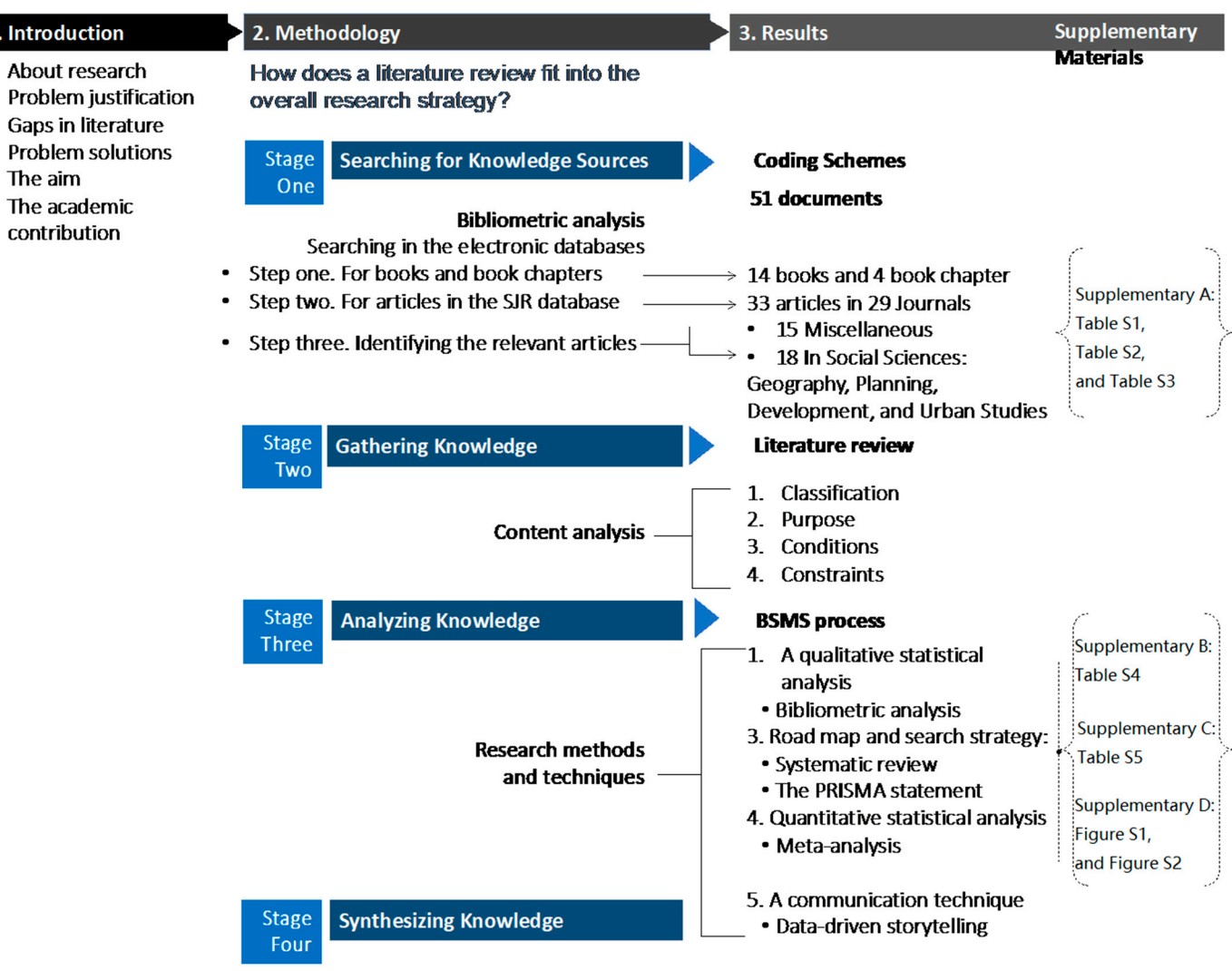

**Figure 1.** Research design.

### 2. Materials and Methods

First, we analyzed how the relevant literature review fits the overall research strategy. We then developed our knowledge of available sources, including books and peer-reviewed journals. If these journals were registered with well-known databases, such as Google Scholar, ResearchGate, Academia, and SCImago (SJR), they must be documented on the Web of Sciences or Scopus. We excluded grey literature. Three tables were created once we accessed the classification method for information sources (Supplementary A–C). In addition, we included reviews, analyses, and syntheses of information from various sources. We depicted four stages of our research design. The first stage involved searching for sources of knowledge—we used bibliometric analysis for this.

Step 1 of Stage 1 identified relevant books and book chapters in electronic databases using four groups of keywords (Table 1), including 18 documents in 14 books and four chapters (Supplementary A). The inclusion criteria of the sources were: (i) published work should be written in English; (ii) source is published between 2005 and 2021; (iii) only publications of Routledge, Sage, and John Wiley and Sons were included; and (iv) the authors of the sources are expected to have an h-index (the number of quotes on Scopus or Google Scholar).

**Table 1.** The keywords for searching on the relevant books and book chapters.

| Group 1 | Group 2 | Group 3 | Group 4 |
| --- | --- | --- | --- |
| "bibliometric", "content analyses", "methods", "research methods", "qualitative, research techniques" | "ANOVA", "validity", "reliability", "clustering", "knowledge" | "systematic literature review", "systematic review" | "urban design", "urban planning", "planning", "design" |

In Step 2, we searched for scientific journals through the SCImago database and identified scientific journals related to the following fields: environmental sciences, medicine, library, and information sciences. It involved searching for relevant articles in electronic databases using four groups of keywords (Table 2). This step revealed 14 journals, from which we chose one or two studies. For each journal, the number of studies was included, and we relied on the exclusion and inclusion of words (Supplementary B).

**Table 2.** The keywords for searching within scientific journals.

| Group 1 | Group 2 | Group 3 | Group 4 |
| --- | --- | --- | --- |
| "bibliometric", "content analyses", "bibliometric approach" | "PRISMA" | "literature review", "systematic review", "scoping review", "meta-analysis", "systematic meta-analysis" | "quantitative", "qualitative", "planning", "design", "impact factor" |

Step 3 identified scientific journals relevant to social science subject areas, including geography, planning, development, and urban studies. We started with the journal ranking in 2020, choosing social sciences as the subject and urban studies as the category, and found 252 journals ranging from Q1 to Q4. We only included scientific periodicals in the Q1 category, resulting in 51 scientific periodicals. In the second round, we only retained journals that contained the words "planning" or "design", resulting in the inclusion of 10 scientific journals. We then conducted a random search for the term "literature review", which yielded over 100 papers. We randomly selected from a search of three papers from each journal, utilizing the following keywords: "review", "narrative", "PRISMA", and "systematic", "meta-analyses", "content analysis", and "bibliometric analysis." We also considered research that includes four groups of keywords (Table 3). This allowed us to limit the available sources to 18 academic papers published between 2019 and 2022, ensuring a rapid review (Supplementary C).

**Table 3.** Keywords for searching academic papers published within 2019–2022.

| Group 1 | Group 2 | Group 3 | Group 4 |
| --- | --- | --- | --- |
| "bibliometric", "content analysis", "literature review", "systematic review", "scoping review", "meta-analysis", "snowball sampling" | "planning", "planner", "urban design", "urban planning" | "quantitative", "qualitative", "comparative analysis", "impact factor" | "knowledge", "storytelling", "narrative", "synthesis", |

In Supplementary D, Figures S1 and S2 highlight keywords used to identify journals for the two study groups. These figures show subject area categories, list of journals, publication date, and article titles. In the last three stages, we employed three techniques for gathering, analyzing, and synthesizing knowledge: (1) content analysis to identify the entities, methods, and practices of literature review, whether derived from selected books or academic research published in peer-reviewed journals (Neuendorf 2016); (2) meta-analysis to determine the frequency of search terms, how to use them, and the compatibility or overlap of word meanings and use locations and (3) data-driven storytelling which uses

descriptive and inferential statistics (Ewing and Park 2020). In this step, we monitored methodologies for preparing a literature review to identify essential components and analysis methods.

## 3. Literature Review

Among scientific research methods, the use of academically applied bibliometric analysis (Amirbagheri et al. 2019), content analysis (Neuendorf 2016), and snowball sampling (Dastjerdi et al. 2021) have witnessed a significant increase in several areas of specialization, such as projection-based clustering through self-organization and swarm intelligence (Thrun 2018), and qualitative research methods in the language (Tracy 2020). Systematic reviews are employed as a method in social sciences (Boland et al. 2014), medical sciences (Higgins et al. 2019), and nursing (Purssell and McCrae 2020).

According to Ball (2017), researchers have adopted the quantitative bibliometric technique to review the relevant literature. However, Verweij and Trell (2019) and Tracy (2020) noted that some researchers use qualitative comparative analysis and its multiple methods for systematic and integrated literature reviews. Scholars have also developed quantitative research methods (Ewing and Park 2020) and analysis of variance (ANOVA) methodologies (Stoker et al. 2020), which were utilized to establish the validity and reliability of research studies (Duke et al. 2020). Systematic reviews and meta-analyses have been combined in the field of medical sciences (Liberati et al. 2009; Moher et al. 2009; Noordzij et al. 2011) and urban studies (Liu and Niyogi 2019; Smith et al. 2021).

A group of public health academics used the PRISMA statement/agreement as a set of preferred reporting items for systematic reviews and meta-analyses (Khan et al. 2011; Moher et al. 2009). This has been replicated in academic research on medicine (Fleming et al. 2014; Liberati et al. 2009; Purssell and McCrae 2020) and nursing (Sandelowski et al. 2007) since 2009 and explored in relevant public health literature (Page et al. 2021).

According to a review of healthcare research, the PRISMA statement helps researchers improve report writing related to structured, systematic reviews and meta-analyses. The PRISMA statement is based on a methodology that explains why we performed a review and what we did and discovered transparently (Liberati et al. 2009). Page et al. (2021) developed methods for identifying, selecting, evaluating, and synthesizing studies. This statement explains the sequence of systematic reviews organized across 27 items: (1) Reference title; (2) Summary; (3–4) Introduction (logic and objectives); (5–15) Methodology (eligibility criteria, information sources, search strategy, information base, selection process, information collection process, listing, and definition of the variables for which data were sought or grants, risk assessment of bias, summary of measurements, statistical or meta-analysis synthesis of results, bias assessment, and certainty assessment); (16–22) Outcomes (case selection, characteristics, case study bias, individual cases, outcome synthesis, cross-case bias, and additional analyses); (23) Discussion (summarizing evidence, limitations, and summary); and (24–27) other information.

Academics have used The PRISMA statement for urban research. For example, the study "Urban blue spaces and human health: A systematic review and meta-analysis of quantitative studies" conducted by Smith et al. (2021) consisted of the following elements: title, authors' names, and citation method; the research questions and what the research aims to establish; the information, based on which the study demonstrates the inclusion and exclusion criteria. The field of study includes the participants (i.e., individuals, groups, municipalities), intervention(s) (i.e., focus on survey forms), exposure(s), comparison (i.e., focus on cases that are not selected), and comparator(s)/control, biosphere context, quantitative and qualitative outcome data. The risk of bias (quality) and assessment strategy for data syntheses include narrative synthesis of the available evidence, scoring by reviewers, and method of review (narrative and structured methodology).

Finally, in medical studies, the dates of reporting studies, the termination dates, funding, conflicts of interest, language, country, and acknowledgment of modernization are also included in an open-access online database of systematic review protocols in

PROSPERO (Abusaada and Elshater 2022; Barros et al. 2019; Harris et al. 2022; Smith et al. 2021). Many researchers can benefit from PRISMA 2020 (Moher et al. 2007; Page et al. 2021) to assess the trustworthiness of the findings. In addition, complete reporting allows readers to determine the appropriateness of the methods used. Although PRISMA is used to review literature in urban planning and research studies, the way PROSPERO documents the results is irrelevant to urban planning and research (Abusaada and Elshater 2022).

## 4. Results

In the current study, we explored the differences between research techniques and identified how novice researchers select research techniques that are appropriate for urban planning and design. Literature reviews have two primary goals: criticism and new information. These reviews can follow the scientific study design principles that appear in final review reports. In four stages, the review process can begin with searching for sources of information, followed by knowledge and content analysis, a step-by-step synthesis of analyzing knowledge, and finally, technique and knowledge synthesis.

### 4.1. Stage One: Searching for Sources

First, the bibliometric analysis used Google search, ResearchGate, and Academia to identify the primary areas of debate. This search determined many fields, from business and economics to arts and humanities, to literary criticism of research techniques in public health, physics, social sciences, and humanities, all of which face significant obstacles. For example, when searching for "literature review", "systematic review", and "systematic literature review", Google yielded 1,110,000,000, 291,000,000, and 281,000,000 results, respectively. Academia yielded 4,679,066; 2415; and 1,511,110 research papers, respectively, while the number of documents was incomplete in ResearchGate. Using the Scimago database, a more specialized survey helped narrow the disciplines to miscellaneous topics and categories, which included four independent groups: (a) environmental sciences: miscellaneous; (b) natural sciences, medical research, and engineering: multidisciplinary; (c) social and human sciences: library and information sciences; and (d) geography, planning, development, transportation, and urban studies: social sciences.

Every document should include data on the breadth of research approaches in general, and urban planning and urban design in particular. Results focused on research using the terms "bibliometric analysis", "content and epistemological analysis", "ANOVA", "validity and reliability of this study", and "systematic literature review and narrative review." We also searched for research published in peer-reviewed scientific journals using the last phrases and terms, in addition to those specified by specific areas of study. Therefore, we gathered data starting from the year 2006. When it came to diversifying research resources, this study employed the PRISMA statement and meta-analysis, along with words such as "evidence" and "synthesis." The results produced 51 documents in two groups of coding schemes.

The first coding scheme (Supplementary A) contained 18 documents include 14 books, and four book sections on four topics: (1) digital computers (Heiberger and Neuwirth 2009; Thrun 2018); (2) language and library arts (Ball 2017; Gough et al. 2017; Noordzij et al. 2011; Todeschini and Baccini 2016); (3) public health (Boland et al. 2014; Higgins et al. 2019; Khan et al. 2011; Popay et al. 2006; Purssell and McCrae 2020); and (4) urban planning and design (Cocchia 2014; Ewing and Park 2020; Duke et al. 2020; Leedy and Ormrod 2018; MacCallum et al. 2019; Stoker et al. 2020).

The second coding scheme was based on a search in the Scimago database, which produced 33 academic manuscripts that discuss literature reviews in all forms. These manuscripts are distributed across multiple disciplines and divided into two groups of topics and categories:

1.     Fifteen manuscripts on management control and information systems (Amirbagheri et al. 2019) earth, atmospheric and planetary sciences (Liu and Niyogi 2019), public health (medicine and nursing) (Amirbagheri et al. 2019; Fleming et al. 2014; Garfield

2006; Kastner et al. 2012; Liberati et al. 2009; Moher et al. 2009; Page et al. 2021; Zhang et al. 2021), research methods (Arksey and O'Malley 2005; Wiles et al. 2011), and physics (Hirsch 2010); and

2. Eighteen manuscripts of social sciences, including geography, planning, development, and urban studies, which are distributed as follows: city planning (AlKhaled et al. 2020; Kwon and Silva 2020; McLeod and Schapper 2020; Navarro-Ligero et al. 2019; Xiao and Watson 2019; Zitcer 2017), urban planning (Chapain and Sagot-Duvauroux 2020; Francini et al. 2021; Özogul and Tasan-Kok 2020; Verweij and Trell 2019; Lim et al. 2019; Pelorosso 2020; Smith et al. 2021; Tarachucky et al. 2021), and urban planning and urban design (Dastjerdi et al. 2021).

### 4.2. Stage Two: Knowledge Gathering and Content Analysis

Literature reviews strive to build on the existing research and provide new insights. They investigate current notions, offers creative ideas, develops community frameworks for academics, decision-makers, and implementers, and provide process and tools that may assist in the resolution of fundamental concerns. The classification, purpose, conditions, and limitations describe the goals of the researcher in scientific manuscripts in urban planning and design. The outcomes include four steps.

**Classification:** The review is a stand-alone piece representing the theoretical foundation for an empirical or applied study to identify gaps and define the reasons for choices (Xiao and Watson 2019). It includes two basic types. The first comprises a narrative review, which provides an analytical description of evidence. It depends on the author's experience, knowledge, and ability to present the topic qualitatively. Subjectivity may make it vulnerable to bias (Noordzij et al. 2011). Its analysis styles include "scoping review", (Arksey and O'Malley 2005; Kastner et al. 2012) "narrative synthesis", (Popay et al. 2006) and "meta-summaries" (Sandelowski et al. 2007). The second is a systematic literature review (Boland et al. 2014; Gough et al. 2017; Higgins et al. 2019) and a meta-analysis of quantitative studies (Hughes 2015; Liu and Niyogi 2019; Noordzij et al. 2011), which focus on quantitative and qualitative studies that follow PRISMA standards and statistical evaluations (Fleming et al. 2014; Liberati et al. 2009; Moher et al. 2009; Page et al. 2021).

**Purpose:** Literature reviews aim to reorganize theoretical and empirical research (Boland et al. 2014). Research goals and objectives include identifying research gaps (Xiao and Watson 2019) and areas for future research, providing evidence, formulating analysis, and communicating the expected or inferred effect from the analysis (Khan et al. 2011; Tracy 2020). Researchers can use review articles to stay updated about the latest scientific advancements and locate novel perspectives. When the indicators are consistent with the study's broad and standard description, a theoretical model or conceptual framework may be developed for resolving a particular issue. Conditions: Literature reviews provide an integrated, impartial, and critical investigation without arguments. Creating a single statement for the literature review depends on three factors: (1) obtaining relevant sources of information and proving their reliability; (2) creating a single statement for the literature review question; and (3) making a single account for the literature review question, referring to debates in the literature. This issue establishes a repeatable technique, that is, identifying core ideas, clarifying the purpose, and establishing a repeatable approach. All assertions must be supported by evidence. Moreover, it demonstrates the researcher's theory versus the studied or challenged reality. Relying on ideas that do not align with reality can be dangerous or beneficial for revising one's work.

**Constraints:** Constraints in literature reviews indicate unrecognized obstacles and issues confronting the study. The findings revealed six limitations that restrict a researcher from providing a proper literature review roadmap.

Impartiality is a challenge in editing literature reviews. Selecting sources for review or rejecting new and unpopular data should not be tainted with bias. Bias is defined as a notion of "for or against" a particular topic or author's perspective. Documented evaluation and collection of compelling evidence from previous studies, rather than personal

judgments (Higgins et al. 2019; Moher et al. 2009), is known as the methodologically rigorous Cochrane review. A systematic review of the literature focuses on answers to specific research questions, which are formulated based on relevant information, compiled and reviewed systematically (Higgins et al. 2019), and primarily used in the medical sciences (Moher et al. 2007).

The review subject, target audience, excessive ambition of the researcher, and focus on one issue without ramifications comprise additional challenges. Therefore, literature reviews must be based on unique questions and avoid expanding to other issues, even if they are interrelated. These should clarify the answer within a clear and sequential line of thought. Dispersion is often evident due to the multiplicity of research questions, topics, and concepts (Tracy 2020). The target audience is determined to adapt to the relevant topic according to the field of publication. For example, if the issue targets city designs, the target audience will consist of urban planners and designers (Abusaada and Elshater 2020, 2021a, 2021c; Elshater and Abusaada 2022a, 2022b). If the topic is urban design of free-roaming cats' habitats, published in the Journal of Applied Animal Welfare Science (Abusaada and Elshater 2021b), the target audience will comprise those interested in animal welfare and urban planners and designers.

A third challenge is choosing research methods, techniques, and approaches compatible with the research topic (Boland et al. 2014; MacCallum et al. 2019). This selection should rely on a single review approach and integrate it with other methods, such as systematic reviews and quantitative synthesis (Kastner et al. 2012). Synthesis methods have appeared in public health literature and are used when evidence is heterogeneous, methodologically diverse, difficult to categorize, and contradictory (Kastner et al. 2012). This reduces random errors in the Materials and Methods section (Fleming et al. 2014).

These methods should be valid, reliable, and reproducible (Xiao and Watson 2019). Reviews should extract data and relevant sources that meet the eligibility criteria (Liberati et al. 2009), focusing on indexed databases (Leedy and Ormrod 2018), such as library indexes or Internet databases. The ability to "retrieve Boolean queries" is an accepted criterion (Higgins et al. 2019). These recalls are based on a data-driven approach and input closely related clusters to explain convergence among information, which is known as cluster analysis (Thrun 2018).

The systematic review value depends on what has been accomplished and reports' clarity (Moher et al. 2009). Reviews include three types of titles: (a) informative titles that make basic information easily accessible, such as participants, interventions, comparators, outcomes, and study design, known as the PICOS approach; (b) indicative titles; and (c) declarative titles that provide the main conclusion (Liberati et al. 2009). Commitment to technical writing (Ewing and Park 2020) expresses the writing style as a unique language that reflects the scientific and cultural knowledge base. Those unfamiliar with scientific writing should seek advice from those who can express their thoughts clearly, using logical and consistent language.

### 4.3. Stage Three: Analyzing Knowledge-Research Methods and Techniques

Theorem-type environments (including propositions, lemmas, corollaries, etc.) can be formatted as follows:

**Qualitative statistical analysis:** To track the temporal development of a specific field of knowledge, the bibliometric analysis relies on quantitative measurements of academic publications (Ball 2017; Todeschini and Baccini 2016). The results outline a comparison between publications and indicators, such as year of publication (recentness), type of publication, grey literature or journal (reliability), views and downloads (prevalence), indexing database, the publication's impact factor (Garfield 2006), and the number of citations (Amirbagheri et al. 2019).

This analysis included three aspects: (1) use of query keywords for information sources; (2) use of bibliometric analysis to explore the co-occurrence of keywords, expressed by the green supply chain, which appears in a graphical abstract using VOSviewer to create a

relationship between words based on similarities (Amirbagheri et al. 2019; Van Eck and Waltman 2010); and (3) a search for information sources in electronic databases using keywords and Boolean operators. "Or" and "And" are used when searching for two different words, and "Not" is used to exclude a word. The following examples demonstrate searching for Elsevier data. A single-keyword systematic review across all subject areas and categories yielded 20,705 results, including 3994 web pages, 14,391 books, 2071 journals, and 249 links. After including only social sciences, the number of journals decreased to 189. After considering only geographical planning and development, the number of journals included was 36. Following this, we chose journals most relevant to urban planning, including cities, progress in planning, urban climate, culture, and society.

The second is the search for the following keywords: systematic review OR systematic literature review OR literature review AND scoping review AND meta-analysis AND planning AND urban planning AND planning literature AND research AND storytelling AND narratives OR planners' narratives AND synthesis. The results comprised 59,407 sources, including 9051 web pages, 45,737 books, 2889 journals, and 1730 links.

After including only social sciences, geographical planning and development, urban and regional planning sub-categories, 10 journals were included. After excluding journals with no impact factor, seven remained. Attempts to empirically search for keywords continued. After identifying relevant journals, each journal was searched using the previous keywords. Keywords were classified into clustering groups based on conceptual affinities. The inductive technique uses clustering (Thrun 2018) or cluster analysis to generate abductive inferences that are not preceded by bias or prior knowledge. It uses a CS encoding scheme to demonstrate the encoding, class, and subclasses to which it belongs (Verweij and Trell 2019). Comparisons between books or articles depend on the year and reliability of publication, number of views, information base, impact factor, and number of citations.

**Roadmap and search strategy:** A systematic review and meta-analysis were used. Each method can be used individually and combined into a single consideration. These methods follow an organized, explicit, and reproducible process, first defining the study field, objective, and audience, limiting the data through a specific research question. Following this, we applied Boolean query for the sources, including only relevant sources of information, culminating in the extraction of evidence and synthesis in a new cognitive context.

The narrative review employs language of novel and sequential speech to explain and interpret contents and relies on qualitative analyses of narratives of the claims presented in studies (Wiles et al. 2011). Xiao and Watson (2019) presented two perspectives regarding narrative reviews. First, many academics (Noordzij et al. 2011; Kastner et al. 2012) perceive narrative review as a descriptive overview without critical appraisal or asking specific questions, addressing broader topics. The application follows informal (not standard or systematic) data mining processes, which is an anecdotal overrun of the evidence, and is subject to bias. Second, this analysis may combine narrative and organized methodologies.

Three patterns follow this type of analysis. The "scoping review", presented by Arksey and O'Malley (2005) for evidence synthesis aimed to determine the research scope and nature of the research activity, and the necessity for a full systematic review, summarizing and disseminating results and identifying gaps in existing literature without assessing its value. Kastner et al. (2012) applied it in the medical sciences to demonstrate methods, processes, and knowledge synthesis. The "narrative synthesis", presented by Popay et al. (2006), appears in the PRISMA Statement (Smith et al. 2021). The "metasummary" of quantitative meta-analysis, designed by Sandelowski et al. (2007), follows a systematic approach to summarize findings as sentences and combine them into objective paragraphs placed into a narrative together with quantitative elements, which determine the intensity of effect sizes by dividing the recurrences of results by number of studies.

A systematic review benefits from scattered knowledge with relatively limited empirical evidence (Lim et al. 2019; Purssell and McCrae 2020). It establishes a link between the topic and other studies (Liberati et al. 2009). For example, Francini et al. (2021) asked "What

are the main lines of research in intelligent mobility?" and "Is it possible to define intelligent mobility that includes these aspects?" They chose the query words "intelligent mobility", "intelligent mobility system", "intelligent transportation", and "intelligent transportation system."

**Quantitative statistical analysis:** A meta-analysis is a quantitative synthesis in systematic reviews that integrates the results of relevant studies (Moher et al. 2009). Statistical techniques are used in the Quality of Reporting of Meta-analyses (QUOROM) Statement, which provides more accurate estimates of public health impacts than individual studies included in structured systematic reviews (Liberati et al. 2009). Meta-analysis consists of two steps. The first step consists of the assessment of consistency (AoC) or homogeneity analysis, used while comparing digital image results. Numerical results were collected and compared to determine the expected degree of error (Liberati et al. 2009; Liu and Niyogi 2019). Second, we applied the random effect summary model (Liu and Niyogi 2019), which depends on the ANOVA (Stoker et al. 2020). The results are compared according to correlation coefficients between independent and dependent variables and affected by systematic factors with effect and random elements without effect. The meta-analysis is based on the AoC, which explores the frequencies of search words referring to the three literature review approaches and two source collection and analysis approaches.

**A communication technique:** According to Zitcer (2017), Throgmorton (1996) sees planning's default genre as persuasive data-driven storytelling, which relies on rhetoric and storytelling rather than science and experts. This hypothetical planning requires planners to write future-oriented texts that use language and speech designed to convince a particular audience of their vision's correctness. People become characters and joint authors and are the only ones who can recognize and test the accuracy of the stories. It depends on coherence, fidelity, the narrator's persuasiveness, construction of rhythmic, and imagistic language, employment, characterization, and description. This type of analysis is based on listening.

According to Sandercock (2003), the ability to provide space to hear others' stories in multicultural contexts is more important than the ability to tell them. Sandercock noted that other people's stories could change reality, particularly when they break out of the local realm and bring a new image that can inspire others to create innovative alternatives. Each story contains ideas and leads to inspiration; convincing stories must fit the needs and situation. Sandercock and Attili (2010) innovated a digital ethnographic approach using blogging, photography, video, and gaming. This approach involves in-depth interviews and captures voices, perceptions, and stories. It captures a plurality of votes and perceptions through an interpretive exercise to understand phenomena.

*4.4. Stage Four: Knowledge Synthesis*

The analysis of 33 papers distributed between 15 types of available research and 18 types of research specialized in urban planning and design revealed that the first group appeared in nine articles, including the research question, while six investigations were absent. The research question arose in the second group of 13 papers and was missing from five documents. Three research papers in the first group used more than one question, while five documents in the second group employed more than one. There were variations in the formulation of each question, including how, which, where, when, who, and in which. Systematic review questions are shown in Supplementary B and C.

There have been many descriptions of knowledge collection including "review of the literature" (Francini et al. 2021; Navarro-Ligero et al. 2019), and "systematic literature review or a systematic review of the literature" (Kwon and Silva 2020). Some results showed that "literature review", "systematic review of the literature", and "systematic review" were used in the same sense (Kwon and Silva 2020; Lim et al. 2019; Özogul and Tasan-Kok 2020). At other times, "systematic review of the literature" was used. To achieve more accurate and rigorous results from "review of the literature" (Xiao and Watson 2019), "review and synthesis of the literature" was used for the same purpose (McLeod and Schapper 2020).

Two papers used the term "scoping review" as an independent approach to literature review (Smith et al. 2021), while "narrative review" was linked to structured systematic reviews (Xiao and Watson 2019). Bibliometric analysis is not widely used in urban planning, as it appears only in Chapain and Sagot-Duvauroux (2020). However, to establish the validity of this finding, we must conduct further research. Further investigation is also required to determine why this type of analysis was not used in planning literature. The use of "bibliometric" is limited, including only two research papers on urban planning and urban design (Dastjerdi et al. 2021). Content analysis was used in three studies and was correlated with a systematic review of the literature (Dastjerdi et al. 2021; Xiao and Watson 2019). The use of "evidence" has been limited in systematic reviews, although the answer to the main question primarily depends on compelling evidence.

Only three studies have reported meta-analyses (Chapain and Sagot-Duvauroux 2020; Smith et al. 2021; Xiao and Watson 2019). PRISMA was only used in two papers, the first of which is closely related to public health research (Smith et al. 2021) and the second is mentioned in the subtitle. The requirements of this agreement have not been proved (Lim et al. 2019), nor has the term appeared in any headline address.

The keywords are among the most frequently used words in query resources in planning, urban planning, and urban design, such as: "keywords" (AlKhaled et al. 2020; Amirbagheri et al. 2019; Van Eck and Waltman 2010; Lim et al. 2019; McLeod and Schapper 2020; Xiao and Watson 2019), "Key words" (Kwon and Silva 2020; Verweij and Trell 2019), "search string" (Pelorosso 2020; Tarachucky et al. 2021), "search terms" (AlKhaled et al. 2020; Zhang et al. 2021), "terms" (Pelorosso 2020), and "search code" (Dastjerdi et al. 2021).

The review of social science sources led to similar conclusions. First, knowledge acquisition, which closes a gap in the development of terminology and research in a specific field of knowledge, identifies uncertainties from various perspectives, highlights the key concepts of specialized practice (quality in general and in consulting work), reviews distinguishing normative factors, provides insights into whether positive or negative, hypothetical, or observable outcomes of a specific field are developed, and establishes the basis of academic research. Second, comparing the methods of conducting and collecting knowledge. This investigates emerging technologies, highlights key characteristics of the research methods and techniques and how they are used, makes them attractive to researchers, suggests ways to produce more responsive texts, and makes planners aware of the complete list of available theories across disciplines. To review how genre fits into storytelling and planning, we explored how modeling tools renew urban planning practices based on performance-based approaches (even if not explicitly announced). Third, synthesizing evidence systematically reviews the evidence and synthesizes applications of specialized practice in any process or organization. Fourth, knowledge gathering reorganizes theoretical and empirical research, identifies distinctive characteristics, develops a conceptual framework for assessment, and assesses the current state of academic engagement.

## 5. Discussion

A previous study showed that a planned and repeatable literature review method is the best way to generate additional information by synthesizing existing knowledge. These evaluations were intended to serve as a framework for determining future research priorities. Although the findings reveal a systematic approach, numerous limitations arise from applying criteria that restrict data examination. Although many manuscripts are referred to as literature reviews, systematic reviews, systematic appraisals, and literature reviews, searching for a constrained scope is challenging.

In examining specialist research articles, the suggestions of Moher et al. (2007) are widely accepted by most scholars. The suggestion aims to review the need to follow the best methods used to search for the best available sources without bias; these should be diverse enough to provide a knowledge space that allows one to arrive at new conclusions and address them in a manner that demonstrates a good understanding of their relationship to the research topic. However, we found that public health research findings are more

structured due to their reliance on the Cochrane library/review/database (Higgins et al. 2019), which does not exist in urban planning and urban design databases. Therefore, we recommend suggesting a database similar to that for public health. The optimal time to launch an urban planning and design database may be achieved by performing an integrated, interdisciplinary study that comprises research teams from diverse disciplines.

However, although systematic reviews rely on an open-ended, clear-cut question, several research papers did not ask any questions on environmental (management) sciences (Amirbagheri et al. 2019), medical sciences (Garfield 2006; Liberati et al. 2009; Moher et al. 2009; Noordzij et al. 2011; Page et al. 2021; Zhang et al. 2021), or urban planning (Dastjerdi et al. 2021; Kwon and Silva 2020; Smith et al. 2021; Tarachucky et al. 2021). Other researchers raised several questions in one study (Chapain and Sagot-Duvauroux 2020; Liu and Niyogi 2019).

The number of questions in prior research cannot be estimated. However, our findings have shown that having more than one question may not be an issue, particularly if they are connected and offer a broader picture of the study subject. For issues that go beyond a literature study, the researcher must separate the questions, specify the research topic for the literature review, and explicitly illustrate the methodologies and processes associated in the methodology section.

The theoretical findings reveal that a solid foundation of other theories, concepts, or ideas must underlie the paper's argument. To ensure "originality" and "contribution", the conclusions of the literature study must provide new knowledge. To use this information in future studies, a critical analysis of the data sources classified as information sources and extract conclusions must be conducted. Decent comprehension and rigor are required in the literature review.

Researchers continue to study this issue, but no consensus has been reached regarding the fact that the odd-numbered bibliometric index is better than the h-index (Hirsch 2010). However, there is no evidence of these common errors' validity, except through many researchers' experiences. The similarity of names in Google Scholar cannot be detected as some researchers have citations registered in their names and the name of other researchers who have similar names. Therefore, the error is that the database allows researchers to add any name they see and record their citations. Even if this error is unintentional, it weakens the reliability of the database.

The BSMS process in urban planning and design combines bibliometrics, systematic meta-analysis, and storytelling to help urban planners and designers select appropriate research techniques based on a PRISMA-compliant review. Using Table 4 as an example, the suggested procedure includes five stages and 20 steps.

**Table 4.** The BSMS process.

| Stages | Steps | Techniques |
|---|---|---|
| First. Identifying the gaps in theoretical research | 1. Identifying the scope of investigation: subject area and category.<br>2. Determining a group of words about the proposed research.<br>3. Determining the source types.<br>4. Exploring the repetition of keywords in urban planning and urban design literature. | Snowball sampling |
| Second. Available resources (Logical query) | 5. Choosing the relevant keywords.<br>6. Searching in electronic databases using three logical operators.<br>7. Write down the list of accessible sources. | The green supply chain Using VOSviewer |

**Table 4.** *Cont.*

| Stages | Steps | Techniques |
|---|---|---|
| Third. Coding scheme (Inclusion criteria) | 8.  Criteria for choosing the documents<br>    − Type of the documents<br>    − Language of publication<br>    − Date of publication<br>9.  Criteria for choosing the documents in an electronic database.<br>10.  Criteria for inclusion and exclusion criteria in the study.<br>    − Reliability<br>    − Registration in citation-based indexes<br>    − H index and impact factors<br>    − Reputable<br>11.  Others:<br>    − Subject area and category<br>    − Source data<br>    − Aim and objectives<br>    − Question and sub-questions<br>    − Keywords<br>    − Repetition of keywords<br>    − Methods and techniques<br>    − Perspectives of the authors<br>    − Applications<br>12.  Supplementary Materials | The methodological filter |
| Fourth. Group formation (Exclusion criteria) | 13.  Determining the query words and identify their repetition in the entire text.<br>14.  Grouping query words into categories.<br>15.  Exclude sources that are not included in these groups.<br>16.  Include any sources that are relevant. | Content analysis |
| Fifth. Synthesis of results | 17.  Extracting and placing complete paragraphs into the sources table.<br>18.  Narrative synthesis provides deductive suggestions applicable to the new investigation.<br>19.  Statistical analysis: to quantify how often certain sections are repeated and how much inference can be made from the findings of the new study.<br>20.  Writing readable and understandable research. | Narrative review |

## 6. Conclusions

We conclude the study by arguing that urban planners and designers can select appropriate research techniques by developing more effective processes. The remaining concerns include recharacterizing the methods, patterns, modes, and limitations of knowledge collection and analysis. The first limitation of this study is that methodological technique, which exists in the fields of social sciences and public health, has not been developed in the field of urban planning and design. Its second limitation is its reliance on four methods that do not objectively represent all relevant fields. Therefore, future research should consider the potential impacts of changing societal, economic, and environmental influences more carefully. For example, issues of the city's image in terms of identity and character have not yet been studied using descriptive meta-analysis and access to quantitative comparison results. In addition, wildlife in urban planning and design, such as shelter sustainability patterns for free-roaming cats, have not been studied from quantitative and qualitative perspectives. This study concludes with a general overview of the quality of academic writing. It also includes the selection of methodology, systematic review, or meta-analysis after a complete study of relevant literature, which begins with the identification of its sources and ends with a summary of the results.

The PRISMA statement/agreement is confined to public health. The possibility of creating information centers for reference registration by developing a PRISMA statement/agreement requires further investigation. The present study faced issues related to the third restriction regarding reliability/validity and quality of sources in the literature review. Research should be conducted in more realistic settings to employ the use of current databases with more accuracy, and secondary literature, including high-quality data, may be prepared. However, several researchers are skeptical about its validity.

This study presented a procedure to help researchers in urban planning and design to analyze the literature using mixed methods. The suggested procedure of BSMS differs substantially from PRISMA and should be adapted according to the nature of urban research, unlike the knowledge base covered by PRISMA. Urban planning and design research can report and document evidence in the same manner as the PRISMA statement documents healthcare interventions. This methodology would allow urban planners and designers to confirm the findings of case studies based on a cumulative database of related topics, which facilitates their research. To summarize, this study argues that the PRISMA statement is a suitable approach for medical sciences, and that the BSMS method is a step toward realizing a similar document for acquiring knowledge and documenting evidence in urban planning and design.

**Supplementary Materials:** The following supporting information can be downloaded at: https://www.mdpi.com/article/10.3390/socsci11100471/s1. Supplementary A: Coding scheme. Table S1: Coding scheme in 14 books and four book sections between 2005–2021. Table S2: Coding scheme in environmental science: medicine, nursing, and library and information sciences in 15 articles in 14 journals selected between 2005–2021. Table S3: Coding scheme in urban planning and urban design in 18 articles in 9 journals selected between 2017–2021; Supplementary B: Keywords in environmental sciences. Table S4: Keywords in medicine, interdisciplinarians, nursing, and library and information sciences in 15 articles in 14 journals between 2017–2021; Supplementary C: Keywords in Social Sciences. Table S5: Keywords in development, geography, planning and development, and urban Studies in 18 articles in 9 journals between 2017–2021; Supplementary D: Keywords used to identify the relevant journals. Figure S1: Keywords used to identify multidisciplinary, miscellaneous, and environmental science journals. Figure S2: Keywords used to identify social science journals.

**Author Contributions:** All authors collaborated on the conceptualization, methodology, writing review, and editing of the manuscript. The manuscript has been read and approved by the A.E. and H.A. All authors have read and agreed to the published version of the manuscript.

**Funding:** The study was supported by a grant from The Science, Technology, and Innovation Funding Authority (STDF) under grant number [STDF-BARG 37234].

**Institutional Review Board Statement:** Not applicable.

**Informed Consent Statement:** Not applicable.

**Data Availability Statement:** The following link provides all the data related to this article: https://doi.org/10.6084/m9.figshare.21318036.v2 (accessed on 2 October 2022).

**Acknowledgments:** It is with great appreciation that the authors would like to thank the editors and reviewers for their constructive comments.

**Conflicts of Interest:** The authors declare no conflict of interest. The authors also declare that the research funder had no role in the design of the study, in the collection, analysis, or interpretation of data, in the writing of the manuscript, or in the decision to publish the results.

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
