# Peer review of "Developing Process for Selecting Research Techniques in Urban Planning and Urban Design with a PRISMA-Compliant Review"

_socsci, doi:10.3390/socsci11100471_

Round 1

Reviewer 1 Report

This manuscript shows the potential to contribute to the knowledge of Urban Planning. Despite its potential, the authors need to revise the presentation of the manuscript in a simple way. I find it difficult to follow the paper presentation in the present form. (please see my comments attached).

Author Response

Dear reviewer,

Thank you so much for your valuable feedback and constructive comments. Here we point out our responses to your valuable comments and places of change.

Comments and responses:

Do you mean literature review techniques?

Response: Yes, we meant that, so we modified the text for better readability of the text. Kindly double-check the blue text on Page 1, Lines 35–36. 

what do you mean by changes in research methods?

Response: To improve the readability of the intended meaning, we have clarified the text. Please check Page 1, Lines 35–36.   

Do Moher et al specifically address urban planning? please check your claim 

Response: Indeed, Moher et al (2009) mainly focused on their research on medical studies. So, we have deleted this research. Please check the text on Page 1, Line 37. 

Why is this the case? please explain

Response: Based on your advice to provide the reason for this case, we have changed the text so that we are able to explain why PRISMA does not meet the criteria in urban planning and design that were profound in medicine. 

is there any relevant with urban planning?

Response: In this study, we have added an emphasis on urban design, which forms the main focus of the study. Kindly check Page 2, Line 47.  

The reader expect a more effective process that should be emerged in result/conclusion

Response: In order to make our intended meaning more understandable, we have developed this sentence. 

is this essential ,or make the presentation more complicated?

Response: This diagram aims to identify the keywords used to identify multidisciplinary, miscellaneous, and environmental science journals.

I would suggest that the authors present these research techniques characteristics, strengths and weaknesses in a concise manner (use table/figure) for instance.

Response: Thank you so much for your valuable comment. Table 4 in the discussion section illustrates 20 steps in five stages. 

Reviewer 2 Report

This paper attempted to provide a suitable methodology based on the PRISMA to be used in urban planning and urban design studies. The general aspect of the paper shows that it is well-written in English, it is clear, concise, and complete. The paper can be accepted after addressing below comments in its revised format.      

Abstract

*The abstract does not include the main findings of the study. So, please provide the key outcome(s) in the abstract.

Introduction

*Paragraph 3 line 43: The statement “However, PRISMA still does not meet the criteria for its use in urban planning and design research.” need to be supported by recent reference(s).

*The authors are advised to add one paragraph (after the last paragraph) explaining the structure/organization of the paper.

Material and methods

*The quality of Fig. 2 needs to be improved.

*Paragraph 4 line 94: Random selection of three papers from each journal put very high level of limitation to this study. Consequently, the derived outcomes can be questionable, and it might not be beneficial to the body of knowledge. Therefore, authors are advised to eliminate this limitation so that more insightful results can be obtained.  

Literature review

*The quality of Fig.3 needs to be improved.

*Paragraph 3 line 141: It is not clear “The National Institute for Health Research” is referred to which country. Please also add information regarding international organizations that accept PRISMA-based research, if any.

*Paragraph 3 line 143 and paragraph 5 line 158: Only one reference has been mentioned for the PRISMA-based research in the field of urban research. It is very important that the authors comprehensively explain the relevant studies published in recent years (e.g., in the last decade).  

*Paragraph 6: The problem statement is not explained based on previous research and only 1 reference, which is quite old, is mentioned. Thus, the authors are required to justify and discuss the how the PRISMA method does not meet the criteria for its use in urban planning and design research as well as its quality inconsistency. Also, providing most recent references is essential.  

Conclusion

*It would be better if the authors can consider a greater number of relevant and earlier studies so that the contributions from the study can be improved.

Appendix C

*It seems three references (numbers 12, 14, and 17) are missed in the provided table. Please clarify this.

Author Response

Dear Reviewer, 

Thank you very much for your valuable comments. The following is a list of our responses point by point. Changes in blue are also reflected throughout the entire text of our manuscript. 

Regards,

Comments and our responses

This paper attempted to provide a suitable methodology based on the PRISMA to be used in urban planning and urban design studies. The general aspect of the paper shows that it is well-written in English, it is clear, concise, and complete. The paper can be accepted after addressing below comments in its revised format.      

Response: Thank you for taking the time to review our manuscript. 

Abstract

*The abstract does not include the main findings of the study. So, please provide the key outcome(s) in the abstract

Response: Thank you for your valuable comment. We have added a sentence in the abstract that reflect the research's main findings.

Introduction

*Paragraph 3 line 43: The statement “However, PRISMA still does not meet the criteria for its use in urban planning and design research.” need to be supported by recent reference(s).

Response: We have made some changes to this sentence based on your recommendation for better readability. 

*The authors are advised to add one paragraph (after the last paragraph) explaining the structure/organization of the paper.

Response: We have added one paragraph that shows the organization of our research. I would appreciate it if you could double-check Page 2, Line 86-70. 

Literature review

*The quality of Fig.3 needs to be improved.

Response: Figure 3 has been replaced with better resolution. 

*Paragraph 3 line 141: It is not clear “The National Institute for Health Research” is referred to which country. Please also add information regarding international organizations that accept PRISMA-based research, if any.

Response: We have added the country of The National Institute for Health Research. Please check Page 5, Line 151. 

*Paragraph 3 line 143 and paragraph 5 line 158: Only one reference has been mentioned for the PRISMA-based research in the field of urban research. It is very important that the authors comprehensively explain the relevant studies published in recent years (e.g., in the last decade).  

Response: Thank you so much for your valuable comment. We have added supportive references. Kindly checks and let us know if further modifications are required in this context.  

*Paragraph 6: The problem statement is not explained based on previous research and only 1 reference, which is quite old, is mentioned. Thus, the authors are required to justify and discuss the how the PRISMA method does not meet the criteria for its use in urban planning and design research as well as its quality inconsistency. Also, providing most recent references is essential. 

Response: The problem statement has been modified. Please check the blue text on Page 2, Lines 46–56. 

Conclusion

*It would be better if the authors can consider a greater number of relevant and earlier studies so that the contributions from the study can be improved.

Response: We have added additional recent references. Kindly check the entire text. 

Appendix C

*It seems three references (numbers 12, 14, and 17) are missed in the provided table. Please clarify this.

Response: We have double-checked the references in Appendix C; they are properly positioned.

Round 2

Reviewer 1 Report

Dear Author (s)

Some improvements have been made by the author. I strongly recommend that author pays attention to fulfilling reader expectations according to what the author state in the introduction. For instance, evaluating  4 research techniques, but this has not been presented in a concise manner (e.g strength-weakness in urban planning discipline). I also found some English that needs to be improved. Please see my comment.

Author Response

Dear Reviewer, 

Thank you so much for your precise reading of the entire text. We also appreciate your time and effort in providing us with these helpful comments. 

Below we provide our responses and positions of change in the entire text. Kindly double-check and let us know if further modifications are required, and we will do them all.

Reviewer's comments and our responses 

What happen when the full specification and protocol have not been applied. This will strengthen your argument

  • Response: Thank you so much for your valuable feedback and recommendation. We have provided explanations based on your recommendation. Kindly double-check the blue text on Page 2, Lines 51–54.

2 word according here, author can also express in two sentences

  • Response: We have rewritten this sentence for better readability. Thank you. 

yes of course, but urban planning research just adopt the principle of PRISMA in systematizing the review of the literature.

  • Response: Yes, we believe that urban planning and research follow PRISMA with a limitation on documenting the results in a medical database. In this regard, we have added a similar meaning on Page 2, Lines 53–54. 

Again, if author promised to examine these techniques, the reader expect this point in the result. To simply express this, I suggest (as I wrote in the first review), to present it concisely e.g in table or figure

  • Response: Thank you for this comment. In the results section, we have Table 4, which shows the BSMS techniques as results of our investigation as promised in the introduction. Kindle, double check Table 4. 

Okay, this guide the readers 

  • Response: Thank you for your double-check. 

I suggest put it in the appendices

  • Response: We have added them in the appendices. Kindly check and let us know if further modification is required. 

again, who claim this? why? is there any evidence

  • Response: Thank you for this comment. We have modified the entire text based on your recommendation. Kindly check Page 5, Line 184–192. We also added supportive evidence. 

is it necessary?

Also please check your grammar

  • Response: We have deleted this sentence from the entire text. Kindly double-check. In addition, we have corrected the grammar typo. 

specify the problem and how this is relevant in problematizing the present study 

  • Response: We have modified the text in this paragraph to better the research problem's readability. Please check the blue text on Page 5, lines 190–192.